# Integrated Proteomics and Metabolomic Analyses of Plasma Injury Biomarkers in a Serious Brain Trauma Model in Rats

**DOI:** 10.3390/ijms20040922

**Published:** 2019-02-20

**Authors:** Tao Song, Ying Zhu, Peng Zhang, Minzhu Zhao, Dezhang Zhao, Shijia Ding, Shisheng Zhu, Jianbo Li

**Affiliations:** 1Department of Forensic Medicine, Faculty of Basic Medical Sciences, Chongqing Medical University, Chongqing 400016, China; Song_xinan@163.com (T.S.); 102675@cqmu.cn (Y.Z.); zhaominzhu@cqmu.edu.cn (M.Z.); 2Department of Forensic Medicine, Hainan Medical University, Haikou 571199, China; zhangpeng19881220@163.com; 3College of Pharmacy, Chongqing Medical University, Chongqing 400016, China; zdzcqmu@cqmu.edu.cn; 4Key Laboratory of Clinical Laboratory Diagnostics (Ministry of Education), College of Laboratory Medicine, Chongqing Medical University, Chongqing 400016, China; dingshijia@163.com; 5Faculty of Medical Technology, Chongqing Medical and Pharmaceutical College, Chongqing 401331, China; 6Chongqing Engineering Research Center of Pharmaceutical Sciences, Chongqing 401331, China

**Keywords:** diffuse axonal injury, metabolomics, proteomics, biomarkers, rats

## Abstract

Diffuse axonal injury (DAI) is a prevalent and serious brain injury with significant morbidity and disability. However, the underlying pathogenesis of DAI remains largely unclear, and there are still no objective laboratory-based tests available for clinicians to make an early diagnosis of DAI. An integrated analysis of metabolomic data and proteomic data may be useful to identify all of the molecular mechanisms of DAI and novel potential biomarkers. Therefore, we established a rat model of DAI, and applied an integrated UPLC-Q-TOF/MS-based metabolomics and isobaric tag for relative and absolute quantitation (iTRAQ)-based proteomic analysis to obtain unbiased profiling data. Differential analysis identified 34 metabolites and 43 proteins in rat plasma of the injury group. Two metabolites (acetone and 4-Hydroxybenzaldehyde) and two proteins (Alpha-1-antiproteinase and Alpha-1-acid glycoprotein) were identified as potential biomarkers for DAI, and all may play important roles in the pathogenesis of DAI. Our study demonstrated the feasibility of integrated metabolomics and proteomics method to uncover the underlying molecular mechanisms of DAI, and may help provide clinicians with some novel diagnostic biomarkers and therapeutic targets.

## 1. Introduction

Diffuse axonal injury (DAI) is one of the most serious and complex types of brain injury and is associated with high mortality and poor prognosis [1,2]. Epidemiological studies have found that 70% patients with traumatic brain injury (TBI) suffer from DAI [3,4]. The underlying pathophysiological mechanisms of widespread axonal injury in DAI are quite complex and remain largely unclear [5]. Recent studies have demonstrated that DAI is a multi-stage biological process disorder with substantial molecular alterations and pathway dysregulations involved [5,6,7]. At present, there are still no objective and reliable biomarkers for clinicians to enable early diagnosis of DAI, which results in a high rate of underdiagnosis and an increased disability and mortality risk [8,9]. Given these facts, there remains an urgent need for methods of early DAI detection to facilitate its diagnosis.

The ongoing omics-based research, which holds promise as a high-throughput, noninvasive, and cost-effective analysis tool, has been well utilized in the identification of diagnostic biomarkers, development of novel targeted therapeutics, and exploration of pathophysiological mechanisms in various diseases [10]. Among these, metabolomics and proteomics are the most commonly used and powerful analytical technologies. Metabolomics technologies are focused on the analysis of the level changes of low-molecular-weight endogenous metabolites in a cell, biofluids, or other biological samples, while proteomics methods mainly used to identify significantly changed proteins in biological tissue [11].

As metabolites can directly indicate the aberrant physiological status of the body, they are generally considered to be sensitive markers of physiological activity. However, metabolites could be influenced by various factors and have large individual differences [12]. Meanwhile, metabolomic analysis has the limitation that it only reflects a collective “snapshot” of metabolic perturbations [12]. Our preliminary metabolomics study, based on an integrated ^1^H NMR and UPLC-Q-TOF/MS method, has found several potential metabolite biomarkers of DAI [6]. However, some limitations of our previous study were that only one time point was concerned and there was a lack of a dynamic evaluation of the identified metabolites. In contrast to metabolomics, proteomic analysis could identify differentially expressed proteins that directly affect the metabolic processes. However, due to the diversity of biological samples in their physicochemical properties and the complexity of the biochemical regulation at multiple levels, no single “omics” analysis can fully reveal the precise picture of the system [12]. Thus, integration of multi “omics” analysis may help to acquire a more complete picture of the changes at the molecular level involved in diseases, provide a more precise understanding of entire biological mechanisms and may offer some novel and reliable biomarkers. Among these, the combination of metabolomic and proteomic analysis is regarded as a powerful tool and frequently applied in biomarker discovery and pathophysiological research, such as multiple sclerosis [10], severe septic shock [13], and anaplastic large cell lymphoma [14].

Therefore, in this study, a well-established rat model of DAI was applied, and a combination of UPLC-Q-TOF/MS-based metabolomic analysis and isobaric tag for relative and absolute quantitation (iTRAQ)-based proteomic analysis was utilized with the aims of gaining a holistic view of the molecular mechanisms of DAI and identifying early, novel and reliable biomarkers in rat plasma. Our findings not only identified several novel potential biomarkers of DAI, but also will help to gain comprehensive insights into the molecular mechanisms of widespread axonal damage after injury.

## 2. Results

### 2.1. Evaluation of the DAI Rat Model

Bielschowsky silver staining confirmed that our DAI animal models were established. As shown in Figure 1, the axons in corpus callosum and internal capsule of brain samples from the injured animals demonstrated various degrees of axonal disconnection, swellings, and axonal retraction balls (ARBs), while those from the control animals showed no pathologic abnormalities of any kind. Bielschowsky silver staining results confirmed that our DAI animal models were successfully established and appropriate for further UPLC-Q-TOF/MS-based metabolomic analysis and iTRAQ-based proteomic analysis.

### 2.2. Plasma Metabolic Analysis Using UPLC-Q-TOF/MS Technology

In this study, plasma samples from the injury and control groups were analyzed by UPLC-Q-TOF/MS technology to identify potential biomarkers of DAI. Typical total ion current (TIC) chromatograms of plasma samples from the control, 1 day and 3 day groups in both negative and positive ion modes were shown in Appendix A.

Principal component analysis (PCA) analysis was carried out in this study to obtain an initial overview of the data set. The PCA analysis result showed that the plasma samples of the injury group were mainly separated from that of the control group (Appendix A). In addition, orthogonal partial least squares discriminate analysis (OPLS-DA) analysis was carried out for supervised pattern recognition. The OPLS-DA score plots showed that the 1 day group was clearly segregated from the control group (Figure 2A,B; ESI+: R^2^X = 0.725, R^2^Y = 0.999, Q^2^ = 0.96; ESI−: R^2^X = 0.757, R^2^Y = 0.99, Q^2^ = 0.983), and the results also revealed an appreciable separation between the 3 day and control groups (Figure 2C,D; ESI+: R^2^X = 0.309, R^2^Y = 0.987, Q^2^ = 0.836; ESI−: R^2^X = 0.577, R^2^Y = 0.999, Q^2^ = 0.837). Furthermore, the results of permutation tests showed no over-fitting of the UPLC-Q-TOF/MS data indicating that the OPLS-DA model was positive and valid (Appendix A, 1 day group vs. control group, ESI+: R^2^ = (0.0, 0.64), Q^2^ = (0.0, −0.261), ESI−: R^2^ = (0.0, 0.632), Q^2^ = (0.0, −0.436); Appendix A, 3 day group vs. control group, ESI+: R^2^ = (0.0, 0.988), Q^2^ = (0.0, −0.227), ESI−: R^2^ = (0.0, 0.975) Q^2^ = (0.0, −0.113)). The results not only indicated the good predictive ability of the model, but also suggested significant plasma metabolic changes occurred in rats with DAI. Based on the OPLS-DA analysis, 15 variables (nine in positive mode and six in negative mode) with variable importance in the projection (VIP) value > 1 and *p* < 0.05 were identified as potential biomarkers in the 1 day group (Figure 3A,B) and were listed in Appendix A. And based on the OPLS-DA analysis, 21 differential metabolites (13 in positive mode and eight in negative mode) were identified in the 3 day group (Figure 3C,D; Appendix A). The results of the Shapiro–Wilk test showed that the distribution of the data was normal (Appendix A). Through comparative analysis, 34 metabolites were identified differentially changed in at least one time point, and two metabolites, 4-Hydroxybenzaldehyde and acetone, were identified differentially changed in both 1 day and 3 day post-injury compared with the control. Both of the metabolites were identified by authentic standards.

A total of six within-run quality control (QC) samples were used to evaluate the repeatability of the metabolomic analytical system. As shown in Appendix A, the overlaps of the spectral peaks of the QC samples were within slight changes, indicating the UPLC-Q-TOF/MS system has a good stability and repeatability on an overall view. Then eight ions from chromatographic peaks were randomly picked and used to assess the repeatability of the method in detail (Appendix A), and the result indicated that the reproducibility was favorable.

Metabolomic pathways analysis of these significantly changed metabolites was determined with the MetaboAnalyst tool. The result revealed that eighteen metabolic pathways were changed and displayed as circles in Figure 4. Among these, four marked disturbed metabolic pathways, namely alanine, aspartate and glutamate metabolism, glycerophospholipid metabolism, arginine and proline metabolism and beta-alanine metabolism were found to be the most relevant pathways to DAI (Figure 4). Among the metabolites, glutamine belonged to alanine, aspartate, and glutamate metabolism; phosphocholine and phosphatidylcholine belonged to glycerophospholipid metabolism; glutamine and arginine belonged to arginine and proline metabolism; and 3-Aminopropionaldehyde belonged to beta-alanine metabolism.

### 2.3. Plasma Proteomic Analysis Using iTRAQ

Within both the control and injury groups, a total of 374 proteins were identified and quantified with at least one unique peptide and false discovery rate (FDR) < 1%. According to the criteria of fold changes > 1.5 or < 0.67 and *p*-values < 0.05, 43 proteins exhibited significant differential expression in at least one time point of the injury group (Appendix A). Of these, 33 proteins were identified significantly altered in 1 day group, and 36 proteins were identified significantly altered in 3 day group (Figure 5). Gene ontology (GO) enrichment analysis of these differentially changed proteins was performed in terms of their cellular compartment (CC), biological process (BP), and molecular function (MF; Figure 6A–C). The CC analysis revealed that these proteins mostly belonged to the extracellular exosome (83.3%), extracellular space (71.4%), blood microparticle (54.8%), extracellular region (26.2%), and cell (9.5%). For the MF, these proteins were mainly involved in serine-type endopeptidase inhibitor activity (28.6%), endopeptidase inhibitor activity (11.9%), serine-type endopeptidase activity (11.9%), oxygen binding (9.5%), and protease binding (9.5%). The BP analysis indicated that these proteins were major participants in negative regulation of endopeptidase activity (35.7%), acute-phase response (19%), inflammatory response (14.3%), complement activation, classical pathway (11.9%), innate immune response (11.9%), and response to organic cyclic compound (11.9%). For pathway analysis, these proteins were mainly involved in complement and coagulation cascades (34.2%), Staphylococcus aureus infection (14.5%), primary immunodeficiency (12.5%), metabolic pathways (10.4%), phagosome (10%), Wnt signaling pathway (8.8%), Fc gamma R-mediated phagocytosis (8.8%), and B cell receptor signaling pathway (8.8%; Figure 6D).

Cluster analysis revealed changes in abundance in the 43 significantly altered proteins. Heat map data showed that the changes in protein abundance observed in the 1 day group were significantly different from those found in the 3 day group (Figure 7). In addition, STRING analysis showed that several proteins like Alpha-1-antiproteinase (Serpina1), Kallikrein B, plasma 1 (Klkb1), and Alpha-1-acid glycoprotein (AGP) were found to be located at the hub positions of the protein functional interactions networks (Figure 8).

### 2.4. Validation of the iTRAQ-Based Proteomics Results via Western Blot Analysis

Three differentially expressed proteins, Serpina1, Klkb1, and AGP, were selected for western blot analysis based on their high fold-changes and possible biological functions in the development of DAI to validate the iTRAQ-based quantitative proteomics results. The western blot analysis results supported the iTRAQ-based proteomics results by demonstrating a significantly lower expression level of Klkb1 in the 1 day group and a significantly higher expression level in the 3 day group, and a significantly lower expression level of Serpina1 and AGP in both of the injury groups compared with the controls (Figure 9).

## 3. Discussion

Early diagnosis of DAI is of great importance for the early implementation of appropriate therapeutic interventions, which play important roles in preventing or reversing the deterioration of axonal injuries, decreasing the mortality and morbidity rate, and obtaining a good outcome for patients with DAI [8,9]. A number of potential biomarkers of axonal injury following TBI have been reported in previous studies, such as tau protein, amyloid-β (Aβ) peptides and neurofilament light polypeptide [15]. However, there are still no reliable and objective laboratory-based tests for DAI to facilitate its diagnosis. Identifying critical metabolites and proteins as well as the involved pathways of DAI using proteomics and metabolomics approach may offer us a better understanding of the molecular mechanisms of axonal injuries during DAI, and may provide potential diagnostic biomarkers. Over the past few years, our research group has focused on identifying differentially expressed metabolites and proteins and revealing early biomarkers of DAI. Zhang et al. has found four potential metabolite biomarkers for DAI by an integrated ^1^H NMR and UPLC-Q-TOF/MS-based analytical platform, and found two potential protein biomarkers using an iTRAQ-based analytical platform [6,7]. However, due to the diverse physicochemical properties of the biological samples and the complicatedly pathological processes of DAI, which were involved with multiple molecular levels, we still do not obtain a holistic view of the molecular mechanisms or definitive biomarkers for clinical application of DAI by a single “omics” analysis alone. Thus, an integrated analysis of UPLC-Q-TOF/MS-based metabolomics and iTRAQ-based proteomics profiling is promising and may help to obtain a comprehensive picture of plasma after DAI, which could provide further important insights into the pathophysiology of DAI. In the preliminary iTRAQ-based proteomics experiment, three individual plasma samples collected from each of the three groups were equivalently pooled to minimize biological variation [7,16]. To the best of our knowledge, this is the first study addressing this issue using an integrated metabolomics and proteomics approach. 34 significantly changed metabolites and 43 differentially expressed proteins were identified in rat plasma after DAI in our study.

Reliable biomarkers should have the characteristic of stable expression trends in diseases. Through comparative analysis, acetone, and 4-Hydroxybenzaldehyde were identified significantly differentially altered in both 1 day and 3 day post-injury, and may be used as candidate biomarkers of DAI. Among the two potential metabolite biomarkers, acetone was significantly elevated in DAI plasma subjects relative to the control animals. It is generally known that acetone is a product of acetyl-CoA, and can produce ketone bodies be used as a vital source of energy for the brain under special physiological and pathological conditions [17]. The significant up-regulation of acetone may suggest the elevated flux from acetyl-CoA into acetoacetyl-CoA [16]. Combined with the results of remarkable down-regulation of glucose identified in our previous study, significantly increased acetone may imply an impaired tricarboxylic acid (TCA) cycle in mitochondria and reduced ATP synthesis after DAI [17,18]. The present findings are consistent with the results of our previous study [6]. Previous research has demonstrated that ketones have neuroprotective effects in various neurological diseases, such as Parkinson’s disease, Alzheimer’s disease, epilepsy, and TBI [19,20]. Recently, Denihan’s group applied an untargeted metabolomics technology to examine the cord blood metabolome’s response to hypoxic-ischaemic encephalopathy (HIE) and indicated that the acetone level was associated with the severity grades of HIE [21]. Furthermore, a previous study performed by Wang et al. showed that acetone was changed significantly in the cerebrospinal fluid, and may be used as a biomarker of cerebral ischemia-reperfusion injury [22]. In this study, the continuous up-regulation of acetone demonstrates the feasibility of acetone being used as a potential biomarker. Moreover, our findings also suggest that promoting the expression of acetone after DAI may reduce axonal injury and improve the clinical outcome of patients with DAI.

Another selected metabolite biomarker in our study was 4-Hydroxybenzaldehyde. 4-Hydroxybenzaldehyde was significantly decreased in plasma subjects of injury group. Recent studies indicate that 4-Hydroxybenzaldehyde plays a significant role in inhibiting the occurrence of seizure and convulsion through its effects of antioxidation and positive modulation of GABAergic neuromodulation [23]. Although the detailed molecular mechanisms are still unclear, the decreased 4-Hydroxybenzaldehyde post-injury may be attributed to the persistent unconscious state of patients with DAI.

Of the 43 differentially expressed proteins, three were screened out based on the protein–protein interaction (PPI) analysis result and their high fold-changes and biological functions, and all were successfully confirmed in a larger group of plasma samples by western blot analysis. Among the three differentially expressed proteins, Serpina1 and AGP were identified as potential plasma biomarkers for their stable expression trends in the early stage of DAI. Serpina1, an inhibitor of serine protease and neurotrypsin, was significantly decreased in plasma subjects of injury group compared with controls [24]. Serpina1 plays an essential anti-inflammatory role in protecting tissues from proteolytic mechanisms [24]. Recent studies indicate that Serpina1 is associated with various diseases, such as sporadic amyotrophic lateral sclerosis, cancers, schizophrenia, bipolar disorder, and frontotemporal dementia [24,25,26,27]. Furthermore, a recent study by Peng et al. indicated that up-regulation of Serpina1 could suppress agrin-22 production and eventually lead to synaptic dysfunction [28]. In the present study, the significantly down-regulated Serpina1 may play an important role in maintaining axonal stabilization post-injury.

Another identified protein biomarker in this study was AGP, which was also markedly down-regulated in rat plasma of injury group. AGP, also known as orosomucoid 1, is a positive acute phase protein and has been identified to play significant roles in anti-inflammation, immunomodulation, and vascular protection during the acute-phase response [29]. Research has shown that AGP is evaluated in acute-phase response and has been reported as a biomarker of inflammatory diseases [30]. At present, AGP is thought to offer a certain protection against inflammation-induced tissue injury through the inhibition of TNF-α-induced apoptosis of hepatocytes and reduced prostaglandin E2 generation, as well as inhibition of neutrophil activation [29]. Moreover, recent studies indicate that AGP is associated with many other biological processes including maintaining the barrier function of capillary, binding and carrying drugs and mediating the sphingolipid metabolism [31,32]. The previous study indicates that AGP is associated with various diseases, such as cancer, liver diseases, and HIV [32]. One study demonstrated that decreasing the AGP expression after ischemic stroke could aggravate blood-brain barrier disruption [33]. In the present study, the down-regulation of AGP may be involved in the blood-brain barrier damage underlying DAI. In addition, the present findings suggest that improving the AGP expression may inhibit the blood-brain barrier disruption and improve outcomes of patients with DAI.

In research on DAI, our previous plasma proteomics study also found two proteins, glyceraldehyde-3-phosphate dehydrogenase (GAPDH) and hemopexin (Hpx) were significantly changed in rats with DAI [7]. Combining with results of this study, we found that significantly perturbed energy metabolism, inflammatory response, cytoskeletal disruption, and immunomodulation may all participate in the axonal injury in DAI, which was consistent with previous studies [2,15].

There are some limitations within this study. First, because DAI is often associated with multiple other organ injuries in clinical, the specificity of the four candidate biomarkers needs to be confirmed by further experiments. Second, the height of the injury hammer was only set at 1.5 m in this study, which means that we cannot ensure the four candidate biomarkers could be used to establish a dose-response correlation to severity of the injury. Third, all of the findings in our study were based on a DAI rat model. Therefore, future studies are needed to determine whether the four candidate biomarkers can be extended to human patients.

Above all, this study first applied an integrated metabolomics and proteomics approach to investigate the alterations of metabolic and protein profiles in the plasma of rats with DAI. A total of 34 differential metabolites and 43 differential proteins were screened out. Two significantly changed metabolites (4-Hydroxybenzaldehyde and acetone) and two differentially expressed proteins (Serpina1 and AGP) were identified as potential biomarkers for DAI. In conclusion, the results of our study not only offer further insights into the molecular mechanisms of DAI, but also may provide clinicians with some novel diagnostic biomarkers and therapeutic targets. Future studies are required to validate our findings and analyze the acquired metabolomic and proteomic data deeply.

## 4. Material and Methods

### 4.1. Chemicals and Reagents

All formic acid, methanol, and acetonitrile used in this study were of HPLC grade. Formic acid and acetonitrile were purchased from TEDIA (Fairfield, IA, USA). Methanol and authentic standards were obtained from Sigma-Aldrich (St. Louis, MO, USA). iTRAQ Reagents Multi-plex kit was purchased from Applied Biosystems (Foster City, CA, USA). Ultrapure water was prepared using a Milli-Q water purification system (Millipore, Bedford, MA, USA).

### 4.2. Animals and Ethics Statement

A total of 35 adult male Sprague-Dawley rats with initial weights of 250–300 g were supplied by the Chongqing Medical University Laboratory Animal Center (Chongqing, China). All rats were housed under standard laboratory conditions with a 12-h light-dark cycle, relative humidity of 50–60% and a room temperature of 21–24 °C. Standard diet and water were provided ad libitum. Before the beginning of the experiment, the animals were acclimatized to the environment for one week. All animal procedures complied with the Guide for the Care and Use of Laboratory Animals, and were approved by the Ethics Committee of C.Q. Medical University (project identification code: 20170126; 14 March 2017).

### 4.3. Models and Sample Collection

#### 4.3.1. Experimental Model of DAI

DAI was performed on rats using an injury model adapted from Marmarou et al., which was described in our previous studies [7,34]. Briefly, 27 randomly selected rats were subjected to establish the DAI model. Rats were anesthetized via intraperitoneal injections of sodium pentobarbital (100 mg/kg). After achieving anesthesia, the animal’s scalp was shaved, and a midline incision was performed to expose the vertex of the skull. Then, a stainless steel disc was affixed to the animal’s skull at the midline between the bregma and lambdoid sutures using dental acrylic. A hammer (weighing 450 g) was allowed to fall freely from a height of 150 cm directly onto the stainless steel disc fixed to the animal’s skull. Six animals died post-injury, and the mortality rate was 22.2%. The other rats fell into a coma (5.20 ± 1.28 min) immediately post-injury and gradually regained consciousness. 16 survived rats were randomly selected for further study and assigned to two groups of eight rats each, and sacrificed at survival periods of 1 day (*n* = 8) and 3 day (*n* = 8), respectively. The remaining five rats were euthanized by decapitation after achieving the appropriate level of anesthesia with sodium pentobarbital (100 mg/kg, ip). Eight sham animals underwent all the same surgical procedures of the injured rats, but were not subjected to injury.

#### 4.3.2. Groups

(1) Control group (*n* = 8).

(2) Injury group (*n* = 16, with two subgroups): 1 day group (*n* = 8) and 3 day group (*n* = 8).

#### 4.3.3. Sample Collection

The blood samples were collected from the animals into EDTA-coated blood collection tubes and kept on ice for 1 h. Then, plasma was separated from blood cells by centrifugation at 3000 rpm for 15 min at 4 °C, and rapidly frozen by liquid nitrogen and stored at −80 °C until proteomic and metabolic analysis. The whole brain tissue samples from the rats were immediately fixed in 10% neutral buffered formalin, and then used for histological analysis with Bielschowsky silver staining.

### 4.4. Metabolomic Analysis

The procedures for plasma sample preparation and UPLC-Q-TOF/MS analysis were reported in our study previously [6]. Meanwhile, a QC sample was prepared by mixing equal aliquot (20 μL) from each of the 24 plasma samples for monitoring the repeatability and stability of the UPLC-Q-TOF/MS analytical system. Briefly, ten QC samples were injected at the beginning of the analytical batch to ensure that the system had reached equilibrium, and then one QC sample was analyzed periodically between every four samples. The repeatability and reliability of the analytical system were assessed using the method reported in our previous study [6]. Raw data were firstly mean-centered and Pareto-scaled prior to multivariate statistical analyses [35]. The normalized data sets were then analyzed by SIMCA-P software (v14.0, Umetrics AB, Umeå, Sweden) for PCA and OPLS-DA to observe discrimination between the DAI plasma samples and controls [36]. A cross-validation procedure and testing with 300 permutations were performed to avoid over-fitting of the supervised OPLS-DA model. VIP values produced in the OPLS-DA model were applied to find differential metabolites and variables VIP > 1 were further processed for Student’s t-test. Variables with VIP > 1 and *p* < 0.05 were considered to be significant differential metabolites [37]. Identification of the potential metabolite biomarkers was preliminarily carried out by their accurate mass spectra, retention time and retention index, and further by searching online databases, including the Kyoto Encyclopedia of Genes and Genomes (KEGG) (http://www.genome.jp/kegg/), HMDB (http://www.hmdb.ca/) and METLIN (http://metlin.scripps). Then, the putative identifications were verified by comparing the MS/MS fragmentation patterns and retention time with those of authentic standard compounds. Online MetaboAnalyst tool (http://www.metaboanalyst.ca) was used for metabolic pathways analysis of the significantly changed metabolites.

### 4.5. Proteomic Analysis

#### 4.5.1. iTRAQ-Based Proteomic Analysis

Three randomly selected plasma samples from each of the three groups were equivalently pooled, respectively. A total of 200 μL plasma from each of the three groups was used for proteomic screening. The procedures for plasma sample preparation and iTRAQ-based proteomic analysis were reported in our previous study [7]. The three samples were labeled as follows: The control group sample was labeled with iTRAQ tag 113, and the samples from the rats sacrificed at 1 day and 3 day after injury were labeled with iTRAQ tag 116 and iTRAQ tag 118, respectively. The iTRAQ ratios and p-values were calculated using the ABI ProteinPilot software v4.5. The threshold value was set as > 1.5 or < 0.67 fold change and *p*-values < 0.05. The differentially expressed proteins were functionally annotated according to their biological processes, cellular components and molecular functions by GO annotation. Pathway analysis of the significantly altered proteins was performed with the KEGG online database (http://www.genome.jp/kegg/). Heat map of the significantly changed proteins was visualized using the online Clustvis tool (https://biit.cs.ut.ee/clustvis/). PPI network analysis is useful to understand the molecular mechanisms of various diseases from a systemic perspective. In our study, the PPI network was mapped using the online STRING 10.5 tool (https://string-db.org).

#### 4.5.2. Validation of Differentially Expressed Proteins

iTRAQ-based proteomics results were validated by examining the differentially expressed proteins via Western blot analysis. Briefly, the remaining five plasma samples from each of the groups were equivalently pooled, respectively. Total protein concentration was determined using a BCA Protein Assay Kit (Abcam Inc., Cambridge, UK). An equal amount (50 μg) of proteins was separated by 10% SDS-PAGE and then transferred to PVDF membranes. Next, the membranes were blocked using 5% nonfat milk for 1 h, and incubated overnight at 4 °C with the primary polyclonal antibodies of anti-Serpina1 (Bioworld Technology Inc., Louis Park, MO, USA; catalog number: BS5593), anti-Klkb1 (Abcam Inc., Cambridge, UK; catalog number: ab1006), anti-AGP (Abcam Inc., Cambridge, UK; catalog number: ab200732), and anti-β-actin (Abcam Inc., Cambridge, UK; catalog number: ab8226; both at 1:1000 dilution) and followed by secondary antibody of HRP-conjugated goat anti-rabbit IgG (1:10,000 dilution; Abcam Inc, Cambridge, UK). The membrane was washed three times with TBST. The immunoreactive signals were visualized and quantified using an ECL chemiluminescence detection system (Amersham Biosciences Inc., Piscataway, NJ, USA). All experiments were performed in triplicate and repeated three times, and the signals from target protein bands were normalized against the β-actin contents.

### 4.6. Statistical Analysis

The experimental results were all presented as mean ± SD, and the statistical analyses were performed using SPSS 21.0 statistical software (IBM, Armonk, NY, USA). Significant differences among multiple groups were calculated by one-way analysis of variance (ANOVA) followed by Tukey’s Honest Significant Difference test as a post-hoc comparison. A 2-tailed *p*-value less than 0.05 was considered as statistically significant.

## Figures and Tables

**Figure 1 ijms-20-00922-f001:**
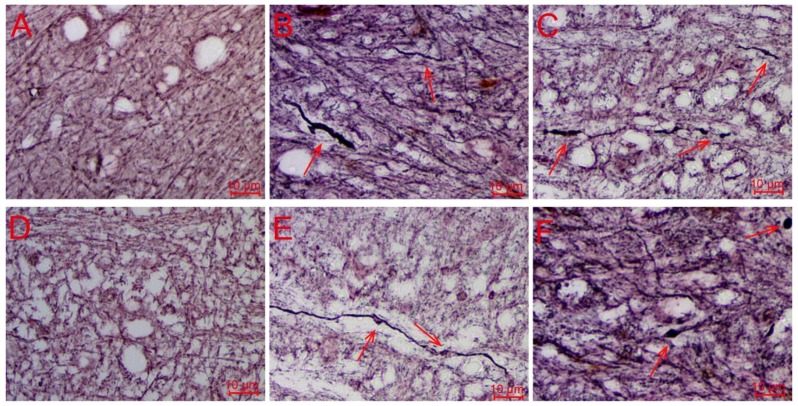
Histopathological examination of the brain tissue samples (200×). (**A**,**D**) control group; (**B**,**E**) 1 day group; (**C**,**F**) 3 day group. The axons in corpus callosum (**A**–**C**) and internal capsule (**D**–**F**) of brain samples from the injured rats demonstrated various degrees of swellings, disconnection, and axonal retraction balls (ARBs), while the control animals demonstrated no pathologic abnormalities of any kind.

**Figure 2 ijms-20-00922-f002:**
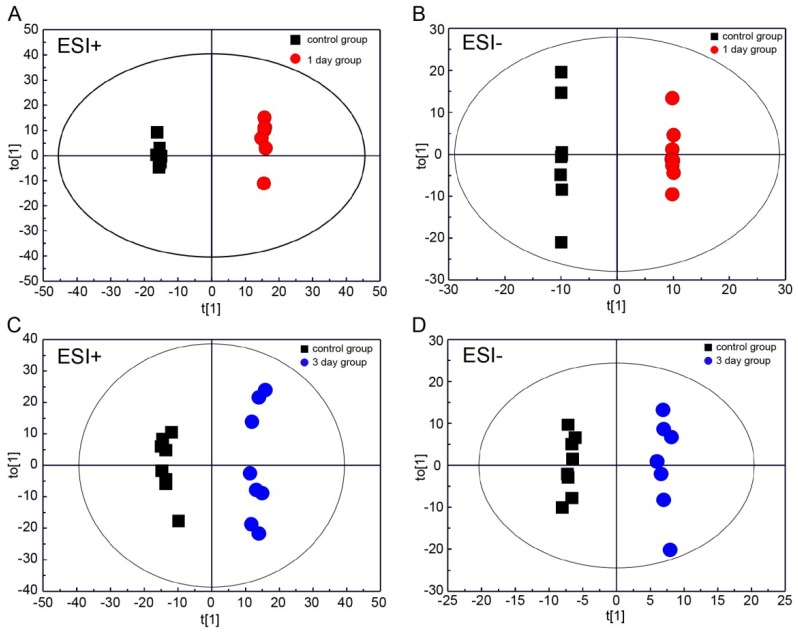
Score plots from OPLS-DA analysis (*n* = 8). (**A**,**B**) control group and 1 day group; (**C**,**D**) control group and 3 day group.

**Figure 3 ijms-20-00922-f003:**
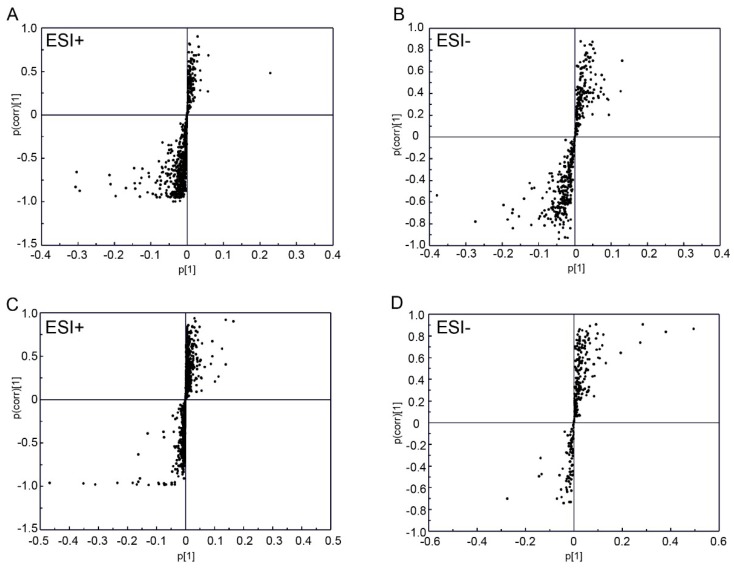
S-plot from OPLS-DA analysis. (**A**,**B**) 1 day group; (**C**,**D**) 3 day group.

**Figure 4 ijms-20-00922-f004:**
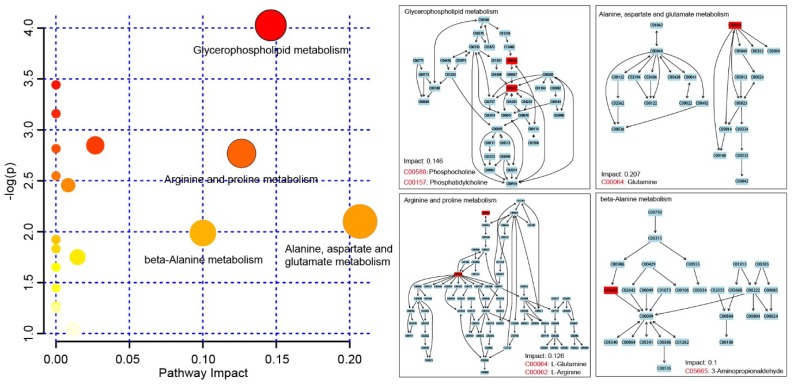
Metabolic pathway analysis of the 34 differentially changed metabolites. The identified metabolic pathways were visualized and arranged according to the score based on enrichment analysis (y-axis) and topology (x-axis).

**Figure 5 ijms-20-00922-f005:**
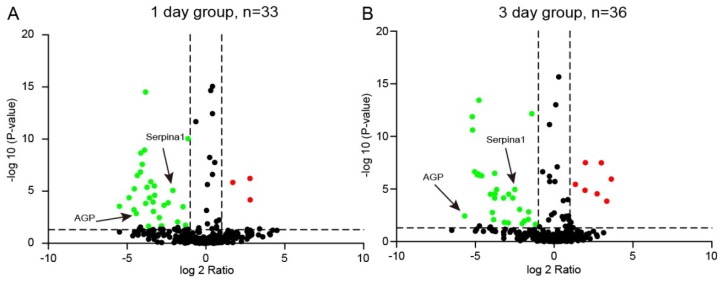
Volcano plot. Red dots indicate represent proteins significantly up-regulated in plasma samples of the injury group, while green dots indicate represent proteins significantly down-regulated in plasma samples of the injury group.

**Figure 6 ijms-20-00922-f006:**
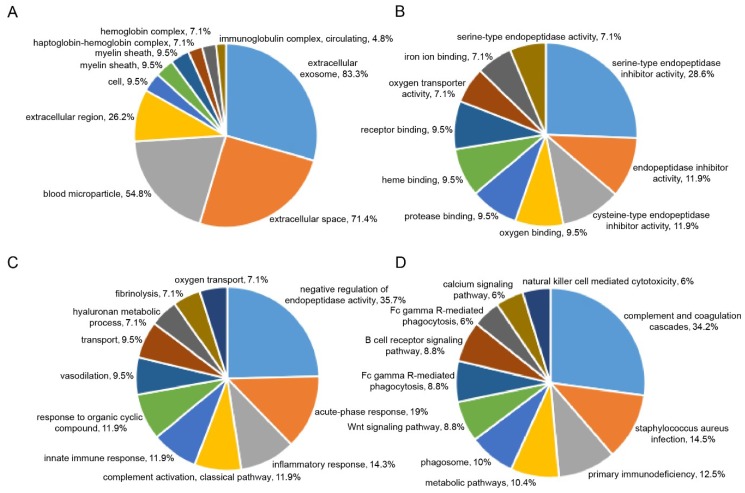
GO annotation analysis and pathway analysis of the differentially expressed proteins. (**A**) cellular compartments; (**B**) biological process; (**C**) molecular function; and (**D**) the most enriched pathways.

**Figure 7 ijms-20-00922-f007:**
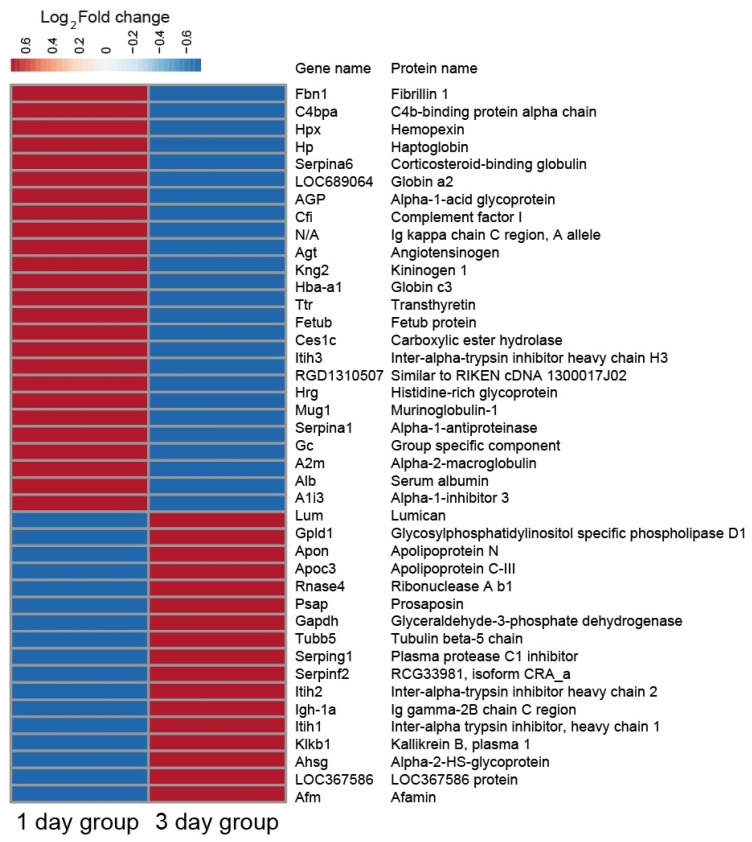
The heat map of the 43 differentially expressed proteins. Red, up-regulation; blue, down-regulation.

**Figure 8 ijms-20-00922-f008:**
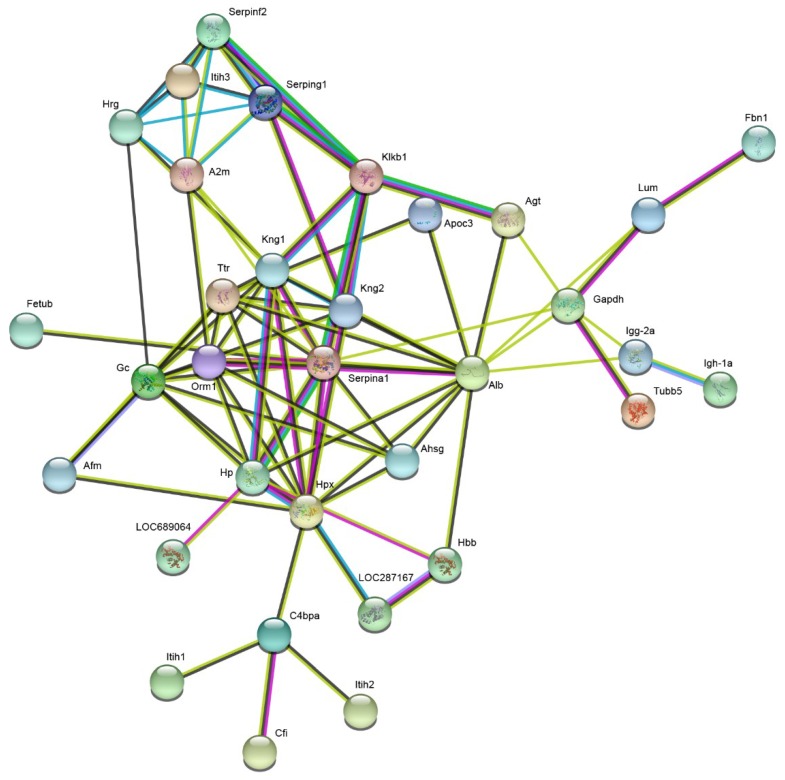
Protein–protein interaction networks of differentially expressed proteins in plasma tissues using the web-based tool STRING v10.5.

**Figure 9 ijms-20-00922-f009:**
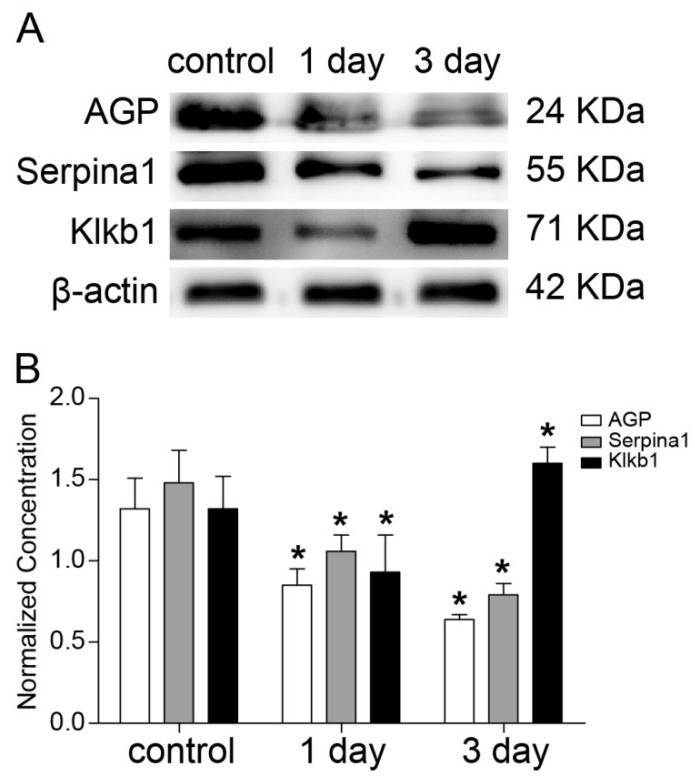
Western blot validation of the three differentially expressed proteins identified by the iTRAQ-based proteomics approach were performed using the remaining five plasma samples from each of the groups. Significant differences among multiple groups were calculated by one-way analysis of variance (ANOVA) followed by Tukey’s Honest Significant Difference test. * means *p* < 0.05 between the control group and 1 day or 3 day group.

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
