# Peer review of "Integrated Proteomics and Metabolomic Analyses of Plasma Injury Biomarkers in a Serious Brain Trauma Model in Rats"

_ijms, 2019, doi:10.3390/ijms20040922_

Reviewer 1 Report

The authors report a proteomic and metabolomic approach to characterizing changes in plasma of rats in a diffuse axonal injury model. Injured animals at day 1 and day 3 after injury were compared to sham animals. Forty-three proteins and 34 metabolites of interest were identified, and further analysis specified metabolites acetone and 4-Hydroxybenzaldehyde and  proteins Serpina1 and AGP as potential biomarkers of diffuse axonal injury. The topic is important, as CT scan imaging may not always conclusively diffuse axonal injury, and MRI may not be able to be obtained early in patients' hospital course, so having biomarkers would be of great clinical utility.. The methodology used is well described. Some English language editing is necessary. Several points are worth further comment by the authors in the discussion, as follows:

  1.  The authors should provide some justification for pooling their samples for analysis, and whether this methodology can lead to errors of interpretation of the results. 

  2.  In humans, diffuse axonal injury often occurs in the context of high-velocity mechanism of injury and is associated with injury to multiple other organ injuries. The authors should comment about the specificity of the changes observed to brain injury, or whether those changes might also be seen in other organ injury. Further study including animal groups with other organ injury, but not head injury, could answer this question. This consideration is important for translation of the authors results to the human condition.

3.  The authors should comment as to whether these markers could be used to establish a dose-response correlation to severity of injury, or whether they likely would be either present or not. 

Author Response

Dear Professor,

Thank you very much for your consideration regarding our above full paper. According to your comments we have revised the manuscript in detail. The revised version of the manuscript with changes in RED has been submitted electronically via the Web. May I reply to your comments and show you the changes in the revision as follows:

1. The authors should provide some justification for pooling their samples for analysis, and whether this methodology can lead to errors of interpretation of the results.

Answer: Thank you very much for your insightful comments. In the preliminary iTRAQ-based proteomics experiment, three individual plasma samples collected from each of the three groups were equivalently pooled to minimize biological variation (J Proteome Res. 2018, 17(1): 499-515; Brain Res Bull. 2018, 142: 224-232). And according to your suggestion, the relevant contents have been added in page 12, lines 222-224. The important related articles have been added as ref. 7 and ref  16.

2. In humans, diffuse axonal injury often occurs in the context of high-velocity mechanism of injury and is associated with injury to multiple other organ injuries. The authors should comment about the specificity of the changes observed to brain injury, or whether those changes might also be seen in other organ injury. Further study including animal groups with other organ injury, but not head injury, could answer this question. This consideration is important for translation of the authors results to the human condition.

Answer: Thank you very much for your good comments. In this study, DAI was induced in animals using an injury model adapted from Marmarou et al., which is a widely accepted and approximate human concussion scenarios animal model (Exp Neurol., 2014, 11, e004). And the injury model established has been successfully confirmed by the subsequent histopathological analysis.

We totally agree with your opinion that DAI is often associated with multiple other organ injuries. And we cannot ensure that the four candidate biomarkers selected are not associated with other organ injury. But, in our study, all four candidate biomarkers were identified not only based on their high fold-changes, but also their biological functions during axonal injury progression, which may significantly contribute to a further understanding of the complex pathophysiological mechanisms of DAI. So, there is a great possibility that the four candidate biomarkers can be used as candidate biomarkers of DAI in rats.

Meanwhile, we agree with you that it is much more better to include animal groups with other organ injury to validate the specificity of the four candidate biomarkers, and we would give more attention to this in our future studies. The relevant contents have been added in page 13, lines 294-296.

3. The authors should comment as to whether these markers could be used to establish a dose-response correlation to severity of injury, or whether they likely would be either present or not.

Answer: Thank you very much for your good and insightful comments. The aim of this study is to explore the candidate biomarkers for early diagnosis of less severe DAI which are not easily diagnosed in clinical. Thus, the height of 1.5 meters was chosen in our study for the reduction in injury intensity and mortality to obtain less severe DAI. And it is a shortcoming of our current study that we did not conduct different severity of injury and cannot ensure the four candidate biomarkers could be used to establish a dose-response correlation to severity of DAI. Meanwhile, we totally agree with you that it is much more better to give more attention to this in our future studies. The relevant contents have been added in page 13, line 296 and page 14, lines 297-298.

4. Some English language editing is necessary.

Answer: Thank you very much for your good comment. The manuscript has been carefully checked for proper English language related to grammar, syntax, word choice, and sentence construction.

Thank you again for your careful review and valuable comments on our manuscript, which would be helpful to not only improve the level of our manuscript but also guide our future research.

Reviewer 2 Report

The author demonstrates a comprehensive proteomic and metabolomic study discovering potential biomarkers of DAI. Following minor isuues need to be addressed:

- The author must mention previously discovered potential biomarkers and methods in other past studies and how their study and methods are better than those

- The study is conducted on rat model. The author needs to emphasize the importance of this study being conducted on human samples and validating of the results using human specimens.

- Change 'metabonomic' to 'metabolomic' on line 125.

- Shortcomings/ improvement scope of the current study should be mentioned in the discussion section.

Author Response

Dear Professor, Thank you very much for your consideration regarding our above full paper. According to your comments we have revised the manuscript in detail. The revised version of the manuscript with changes in RED has been submitted electronically via the Web. May I reply to your comments and show you the changes in the revision as follows:

1. The author must mention previously discovered potential biomarkers and methods in other past studies and how their study and methods are better than those.

Answer: Thank you very much for your good comments. According to your suggestion, previously discovered potential biomarkers of axonal injury following TBI and the advantages of our study has been added in page 11, lines 205-211, and the important related articles have been added as ref. 15.

2. The study is conducted on rat model. The author needs to emphasize the importance of this study being conducted on human samples and validating of the results using human specimens.

Answer: Thank you very much for your insightful comments. According to your suggestion, the relevant contents of the importance of this study being conducted on human samples and validating of the results using human specimens have been added in page 14, lines 298-300.

3. Change 'metabonomic' to 'metabolomic' on line 125.

Answer: Thank you for your careful review. The mistake has been corrected.

4. Shortcomings/ improvement scope of the current study should be mentioned in the discussion section.

Answer: Thank you very much for your good comments. According to your suggestion, shortcomings of the current study have been added in page 13, lines 294-296 and page 14, lines 297-300.

There are some limitations within this study. First, because DAI is often associated with multiple other organ injuries in clinical, the specificity of the four candidate biomarkers needs yet to be confirmed by further experiments. Second, the height of the injury hammer was only set at 1.5 meters in this study, which means that we cannot ensure the four candidate biomarkers could be used to establish a dose-response correlation to severity of injury. Third, all of the findings in our study were based on a DAI rat model. Therefore, future studies are needed to determine whether the four candidate biomarkers can be extended to human patients.

Thank you again for your careful review and valuable comments on our manuscript, which would be helpful to not only improve the level of our manuscript but also guide our future research.